# Public perception of an important urban estuary: Values, attitudes, and policy support in the Biscayne Bay-Miami Social Ecological System

Julia Wester [1,2,3,4] *

**1** Abess Center for Ecosystem Science and Policy, University of Miami, Coral Gables, Florida, United States of America, **2** Department of Anthropology, University of Miami, Coral Gables, Florida, United States of America, **3** Field School, Miami, Florida, United States of America, **4** Field School Foundation, Miami, Florida, United States of America

* Julia.wester@miami.edu

## Abstract

Understanding public perceptions, values, and preferences can be fundamental to effective conservation governance, management, and outreach. This is particularly true in socially and ecologically complex marine and coastal spaces, where many relevant questions remain. The social-ecological system of Biscayne Bay and Miami-Dade are on the frontier of problems that will soon engulf many coastal-urban systems. Despite the economic, ecological, and cultural importance of Biscayne Bay, research into the social components of this social-ecological system is distinctly lacking. In order to effectively address urgent coastal management issues, practitioners and policy-makers need a clear understanding of public perceptions, values, and priorities. In this paper I present the results of a large online survey (n = 1146) exploring public attitudes toward Biscayne Bay as a case study of management and opportunity in a complex coastal social-ecological system. Results describe a public that interacts with and utilizes Biscayne Bay in a variety of ways, from leisure and recreation, to subsistence. This public believes the Bay to be moderately healthy, though somewhat in decline, and has experienced a range of local environmental threats, about which they feel considerable concern. These interactions and concerns are in turn reflected in overwhelming endorsement of value statements regarding the ecological, material, cultural and economic importance of the ecosystem to the city, as well as high levels of support for policy actions to protect and restore that ecosystem. Together these findings indicate that additional policy steps to preserve and restore Biscayne Bay would enjoy support from the local public and demonstrate the power of public perceptions research to identify gaps and opportunities for management and outreach.

## Introduction

Across academic, conservation, and government spaces there have been increasing calls for the integration of social research findings into marine and coastal governance [1]. These

**Data Availability Statement:** Data has been made available via Dryad at the following location: https://doi.org/10.5061/dryad.1rn8pk106.

**Funding:** JW - Save Our Seas Foundational Small Grant, https://saveourseas.com/, The funders had

no role in study design, data collection and analysis, decision to publish, or preparation of the manuscript.

**Competing interests:** The authors have declared that no competing interests exist.

calls have emerged from a growing recognition that while human activities may drive ecological problems, solutions must also necessarily be human-derived [2]. One rapidly growing area of social research is aimed at public perceptions: how does the public view, value, and think about marine ecosystems, related problems and potential solutions [3]. Collectively, the understanding generated by public perceptions research is foundational to effective governance [1, 4, 5].

Many questions at the intersection of public attitudes, social values, and governance remain [4]. Relevant research priorities have been identified in recent years, including the need to determine relationships between various attitudes and values, explore differences within populations, and to investigate patterns at different scales [2, 6]. These complementary strands of public perceptions research can help guide policy development, set priorities, enable behavior change, lend legitimacy and social license to policy actions, and shape effective public outreach and communication strategies [3, 7].

In response to these research and policy needs, the literature on public perception of marine and coastal environments specifically has grown exponentially in the prior two decades [3]. Management of marine environments involves unique complexity which makes public perception even more fundamental: while some coastal habitats are held privately, much are held in the public commons; management often involves overlapping jurisdictions and multiple governing agencies; finally, users may rely on coastal and marine spaces for a wide variety of important services including subsistence, economic support, recreation, and personal and community well-being [3, 4, 8–11].

To date marine public perception research has focused on variety of locations and scales around the globe including but not limited to Ireland, the United Kingdom, Malaysia, Italy, Australia, Taiwan, and Europe as a whole [12–18]. Studies have varied in the social and psychological constructs of interest. Two popular approaches include assessing the knowledge of participants [15, 19, 20] and site-specific user-preferences associated with popular spaces (e.g., beaches) [21–23]. How coastal ecosystems and related services are valued has been investigated using both monetary [24, 25] and non-monetary valuation [26, 27]. Others have measured how the public perceives various coastal environmental threats and risks [6, 7, 28, 29] and how the public understand drivers (causes) and pressures (consequences) of those threats [30, 31].

Significant research has also been dedicated to exploring the relationship between attitudes, values, and perceptions and specific actions. Studies have explored attitudes toward various policies and governance strategies for instance. Jefferson et al., [32] call for making public perceptions research more accessible and applicable for policy-makers and practitioners. Most frequently this kind of research has been directed at public attitudes toward marine protected areas [33–36]. Recently, O'Connor et al., [37] explored within and between group differences in support for ecosystem restoration. Other research has focused on the connection between perceptions and behavioral intentions [28].

Despite the promise and productivity of public perception research for coastal management, findings to date have been anything but simple or straightforward. While often talked about as a monolith, the "public" is a complex, heterogenous group. Social research into public attitudes must not only document population-wide trends, but also should explore variation across groups [4]. Differences have most commonly included demographic factors (e.g., age, gender), personal experience and engagement (e.g., participating in coastal activities), ideological differences, and foundational values or worldviews [3]. These variables can intersect in complex and interesting ways [36, 38]. Thus, studies that explore a wide variety of factors and how they relate to each other can help illuminate patterns and opportunities [6, 39].

It has been argued that conservation efforts and thus research, should be focused in places "where biological richness and human pressures collide most seriously" [40, p 495]. Biodiverse

coastal habitats near urban centers are often the sites of such productive collision. Biscayne Bay for example is a large, shallow, oligotrophic embayment and marine estuary on the southeast coast of Florida, in the shadow of downtown Miami-Dade County, Florida, USA.

Biscayne Bay supports incredible ecosystem and species diversity, and provides the foundation of Miami's economy and cultural identity [41, 42]. The bay contains interconnected ecosystems including mangrove shoreline, seagrass beds, and coral reefs, significant biodiversity including more than 30 endangered species or species of special concern [41], and more than 100 species important to recreational and commercial fisheries [43]. This ecosystem also directly supports approximately 10% of the local economy and more than half a billion dollars in annual tax revenue [42].

The social-ecological system of Biscayne Bay and Miami-Dade is on the frontier of problems that will soon engulf many coastal-urban systems [44, 45]. Human population growth and associated land use change has resulted in shifting stormwater patterns, habitat loss, direct damage from propellers and anchors, and overfishing [41]. Proximity to a dense urban center, sprawling residential communities, and agricultural production has introduced significant nutrient pollution, with resulting eutrophication, algal blooms, and anoxic events [46]. Combined with climate-change-associated trends, such as rising sea levels, warming waters, and more frequent storms, these threats are pushing Biscayne Bay to the brink of ecological collapse [47–49]. In recent years, seagrass die-offs and fish kills indicate the bay is near a tipping point, with potentially devastating and irreversible ecological and economic consequences [42, 49].

The physical, biological, and chemical drivers of change in Biscayne Bay are well documented by the scientific community [46, 50, 51], but meaningful conservation action in the form of social or policy change has been inadequate [52]. A number of institutions are involved in the management of Biscayne Bay resources, including at the County (e.g., Miami-Dade County, Water & Sewer Authority), State (e.g., Florida Wildlife Commission, FWC), and federal level (e.g., NOAA). The statutory guidance and management plans of these often-overlapping jurisdictions have developed through both cooperative governance and judicial action [43, 53].

Attempts to involve community stakeholders have resulted in many reports calling for large-scale action, but little progress with regard to pro-environmental policy change, increased public access to resources, or improved ecosystem health (e.g., Biscayne Bay Partnership Initiative in 2001 and a more recent grand jury report). Public access to Biscayne Bay has decreased, particularly for marginalized communities, because of privatization and waterfront development [53], with environmental and social justice implications [54]. Efforts to solve these problems have been hindered by social factors, including socioeconomic inequality and low civic engagement [55], and ecological factors, including geographic and environmental vulnerability to sea level rise and invasive species [56, 57].

Despite the economic, ecological, and cultural importance of Biscayne Bay, research into the social components of this social-ecological system is distinctly lacking. In order to effectively address urgent coastal management issues, practitioners and policy-makers need a clear understanding of public perceptions, values, and priorities. As this region is at the frontier of so many emergent ecological problems, this case study can further be informative and illustrative for other heavily developed urban-coastal centers. Thus in this paper I present a detailed exploration of public attitudes toward Biscayne Bay as a case study of management and opportunity in a complex coastal social-ecological system. Specifically, I aimed to determine how the public is using and interacting with Biscayne Bay, how the health of the ecosystem is perceived, what values people attribute to the Bay, what threats they have experienced and are most concerned about, and finally what policies actions are supported. I further present data on the

perception of local government and explore how key outcome variables (e.g., values, concern, policy support) are related to individual differences.

## Materials and methods

A sample of Miami-Dade County residents ($n$ = 1146) was recruited using Centiment (centiment.co) from May 10 through December 19 of 2022. Participants were directed to a survey hosted on Qualtrics (qualtrics.com). Participation was limited to adults and was offered in both English and Spanish. At the start of the survey participants were presented with information about the study and provided written consent in the form of checking a box to proceed before they were able to begin the survey. The survey included a combination of Likert-type scales and open-response items which measured perception of the current status of the Bay, as well as changes in ecosystem health over time, relative concern about and personal experience with various local environmental problems, frequency of various recreational uses, perceptions of local governance, and support for a variety of frequently proposed policy actions.

In addition to Likert-type scales, willingness to pay items were used to assess relative support for different approaches to addressing environmental problems in Biscayne Bay. Item design was based on best practices outlined in [58]. Participants were randomly assigned to see one of three potential scenarios regarding septic tank conversion, upgrading water treatment infrastructure, and restoration of natural habitats along the coast. Participants were asked whether they would vote to support the initiative if it would raise their annual taxes. Those that entered "yes" were asked "What is the highest annual dollar ($) amount you would be willing to pay before you would no longer support this policy?". Those that entered "no" were asked to specify why using a checklist of options. Participants could check all options that they felt applied to them and could write in a response if a reason was not represented.

Scales and items were presented in a randomized order. Demographic information including age, race and ethnicity, gender identity, household income, zip code and political ideology and affiliation were collected at the end of the survey. The survey was estimated to take approximately 10 minutes to complete. Those that took less than 2 minutes or that failed to correctly answer an attention check were excluded from final analysis. Data were analyzed using a combination of descriptive and inferential statistics. Where applicable, responses were checked for assumptions of normality. Within-subject difference tests were used to assess relative ratings of problem concern, values, and policy support items. This study was reviewed and approved by the Institutional Review Board at the University of Miami (protocol #20191209).

## Results

### Sample description

Demographic data are reported in Table 1. Respondents ranged in age from 18 to 91 years ($m$ = 40.94, $SD$ = 15.61) and reported having lived in Miami from less than 1 to 80 years ($m$ = 23.99, $SD$ = 16.43). A range of political ideology and affiliation was also represented. Forty percent of respondents identified as a Democrat, 22.2% identified as a Republic and a final 21.3% identified as an Independent. On a five-point scale of political ideology, the median response was three, or "neutral", and responses were normally distributed ($m$ = 3.12, $SD$ = 1.18, skewness = -0.17, kurtosis = -0.79). Responses originated from across the county (participants reported living in 85 different zip codes in Miami-Dade).

**Table 1. Demographic characteristics of the sample.**

| | | n* | % Sample |
|---|---|---|---|
| **Gender** | | | |
| | Female | 573 | 50.0% |
| | Male | 523 | 45.6% |
| | Gender non-conforming | 11 | 1.0% |
| | Not listed/prefer not to answer | 7 | 0.6% |
| **Race** | | | |
| | White | 757 | 66.1% |
| | Black | 228 | 19.9% |
| | Asian | 20 | 1.8% |
| | Am. Indian/Alaska Native | 8 | 0.7% |
| | Native Hawaiian or other Pacific Islander | 4 | 0.4% |
| | Other | 98 | 6.2% |
| **Ethnicity** | | | |
| | Hispanic or Latino | 615 | 53.7% |
| | Not Hispanic or Latino | 496 | 43.3% |
| **Education** | | | |
| | Less than high school | 31 | 2.7% |
| | High school degree | 188 | 16.4% |
| | Some college | 221 | 19.3% |
| | Associate's degree | 124 | 10.8% |
| | Bachelor's degree | 352 | 30.7% |
| | Advanced degree | 199 | 17.4% |
| **Residence** | | | |
| | Own | 295 | 40.0% |
| | Rent | 457 | 41.9% |
| | Live with family/friends | 159 | 13.6% |
| | Primary | 797 | 69.5% |
| | Not primary | 314 | 27.4% |

* Due to missing data the total *n* does not always add up to 1146 for each variable.

## Use and interaction with Biscayne Bay

Participants reported a variety of uses of Biscayne Bay (Table 2). Eighty nine percent of respondents reported using or interacting with Biscayne Bay at least once each year, with the most common and frequent use being going to the beach. Participants individually reported engaging in a range of activities: on average participants reported engaging in at least three of the six

**Table 2. How participants reported interacting with Biscayne Bay.**

| Use | Mean use per year | SD | Median excluding non-users | % who do this at least once a year |
|---|---|---|---|---|
| Going to the beach | 9.82 | 28.17 | 5 | 85.1% |
| Fishing for food | 2.49 | 10.26 | 3 | 69.6% |
| Boating | 3.89 | 15.67 | 3 | 54.0% |
| Scuba diving or snorkeling | 2.23 | 13.96 | 3 | 43.4% |
| Visiting waterfront restaurants | 5.58 | 12.21 | 3 | 38.0% |
| Recreational fishing | 3.53 | 17.74 | 3 | 38.8% |

listed activities at least once each year (*m* = 3.24, SD = 2.05, median = 3), with a plurality of participants (25.2%) saying that they engaged in all six activities at least once per year.

## Perception of ecosystem health and trend

On a 5-point scale ranging from "not at all healthy" to "extremely healthy", the median respondent reported believing Biscayne Bay is currently "moderately healthy" (*m* = 2.81, *SD* = 1.19, skewness = 0.34, kurtosis = -0.55) (Fig 1a). When asked to assess the trend of Biscayne Bay ecosystems over the last ten years on a 5-point scale from "strongly declined" to "strongly improved', participant's median response was 2, or "somewhat declined" (*m* = 2.75, *SD* = 1.32, skewness = 0.42, kurtosis = -.098) (Fig 1b).

## Values associated with the Bay

Overall participants strongly agreed with statements about the value and importance of Biscayne Bay. A statement noting the importance of the bay to the local economy of Miami (*m* = 4.37, *SD* = 0.87) was agreed with marginally but statistically less than statements about the importance of Biscayne Bay as an ecosystem (*m* = 4.50, *SD* = 0.80) and the importance of the bay to Miami's identity and culture (*m* = 4.44, *SD* = 0.85) (Fig 2).

## Environmental threats: Experience and concern

The overwhelming majority of participants (96.5%) reported having observed, experienced or been affected by at least one local environmental threat. The median number of threats experienced was 1 (*m* = 2.83, *SD* = 1.95) (Table 3).

Participants were on average very to extremely concerned about a variety of local environmental issues. When compared, participants were significantly more worried about plastic pollution and less worried about algal blooms than other issues ($F(9.7) = 36.71$, $p < 0.001$) (Fig 3).

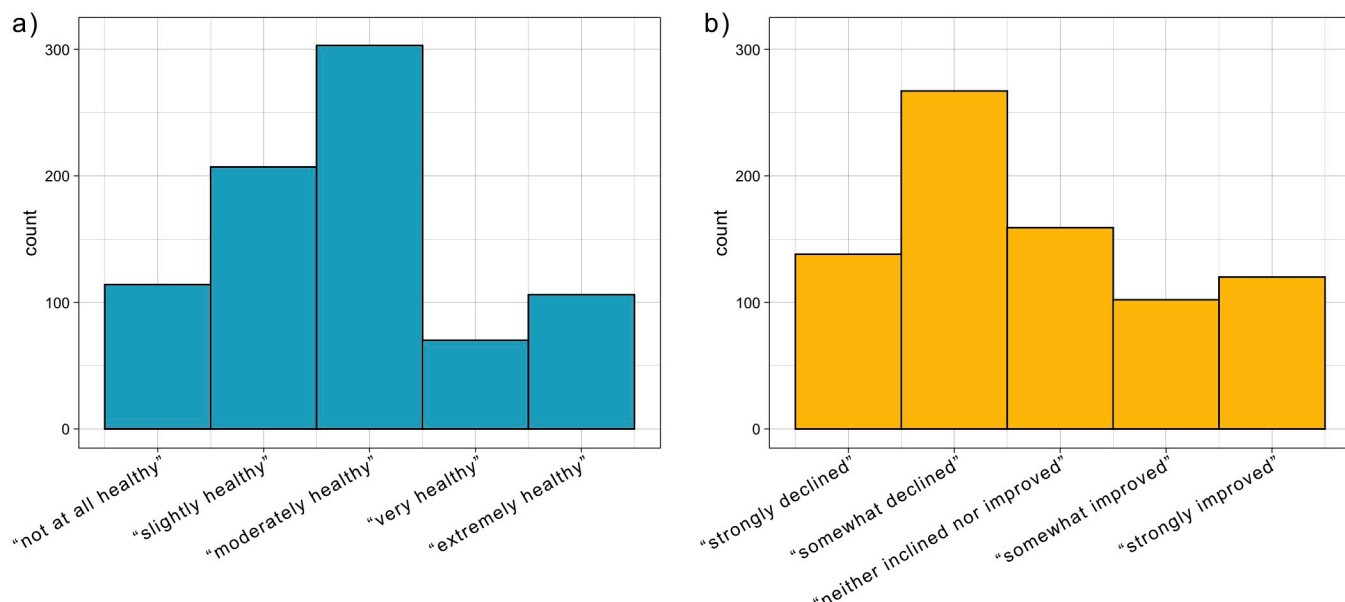

**Fig 1. Perception of Biscayne Bay ecosystem health and trend.** Frequency of responses to the following prompts: a) "The ecosystems of Biscayne Bay are currently. . ." and b) "In the last ten years, the health of Biscayne Bay has. . .".

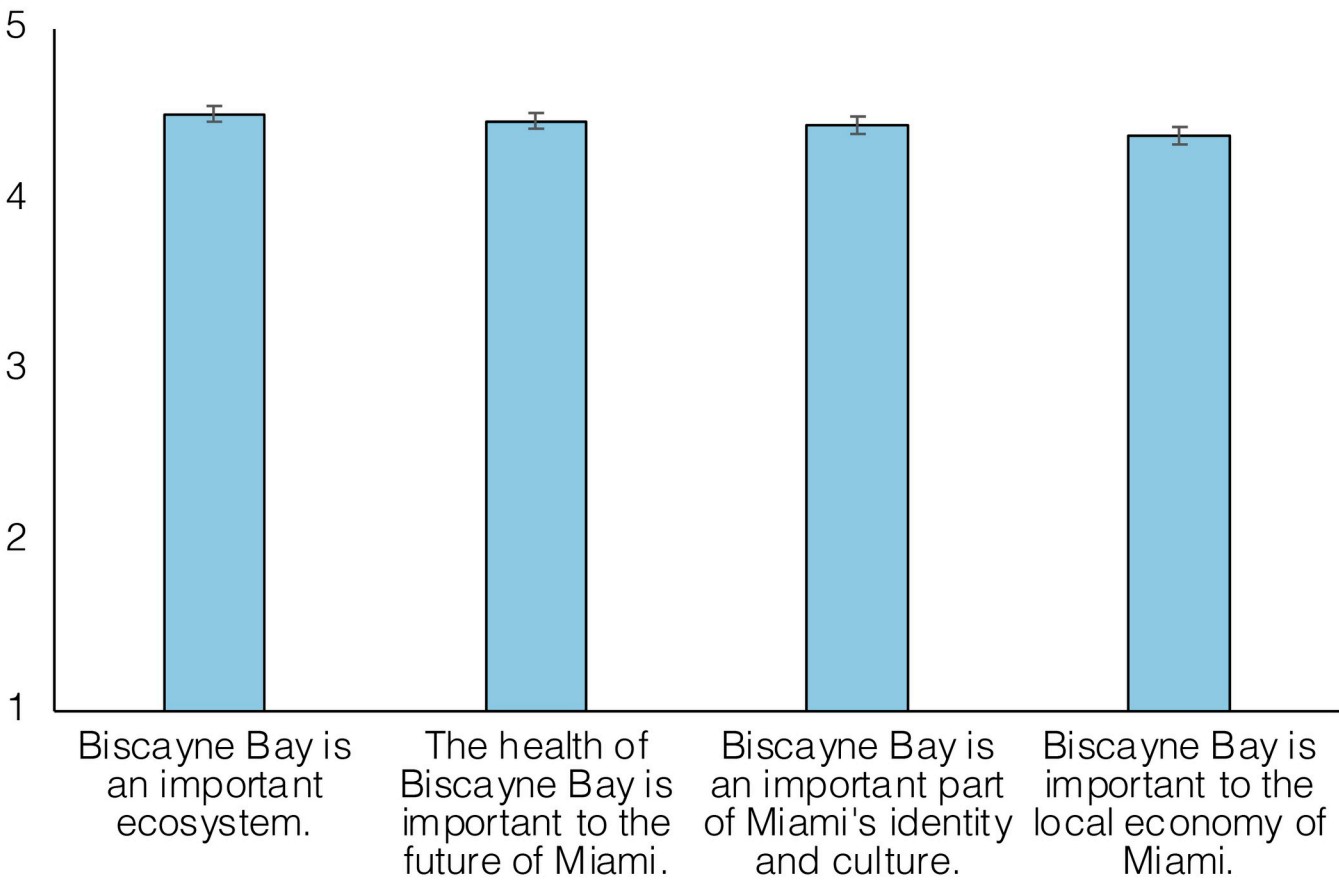

**Fig 2. Values attributed to Biscayne Bay.** Estimated marginal mean response to value statements associated with Biscayne Bay on a scale from 1 ("strongly disagree") to 5 ("strongly agree"). Error bars represent 95% confidence interval.

## Policy support

When policy support was compared, participants were significantly more supportive of policies that would restore local ecosystems and invest in waste water management, and were less supportive of regulating commercial and recreational fishing and of replacing seawalls with natural shorelines ($F(11.1) = 26.61$, $p < 0.001$) (Fig 4). Average support for all policies,

**Table 3. Percentage of participants who reported having personally observed, experienced or been affected by each threat.**

| Threat | % |
|---|---|
| Plastic pollution along the coast | 59.1 |
| Damage from storm surge or flooding | 39.9 |
| Degraded or worsened water quality | 37.9 |
| Fish population declines | 32.3 |
| Damaged or degraded habitats in Biscayne Bay | 30.1 |
| Fish die-offs | 28.3 |
| Coastal sewage spill | 23.0 |
| Harmful algal blooms | 19.7 |

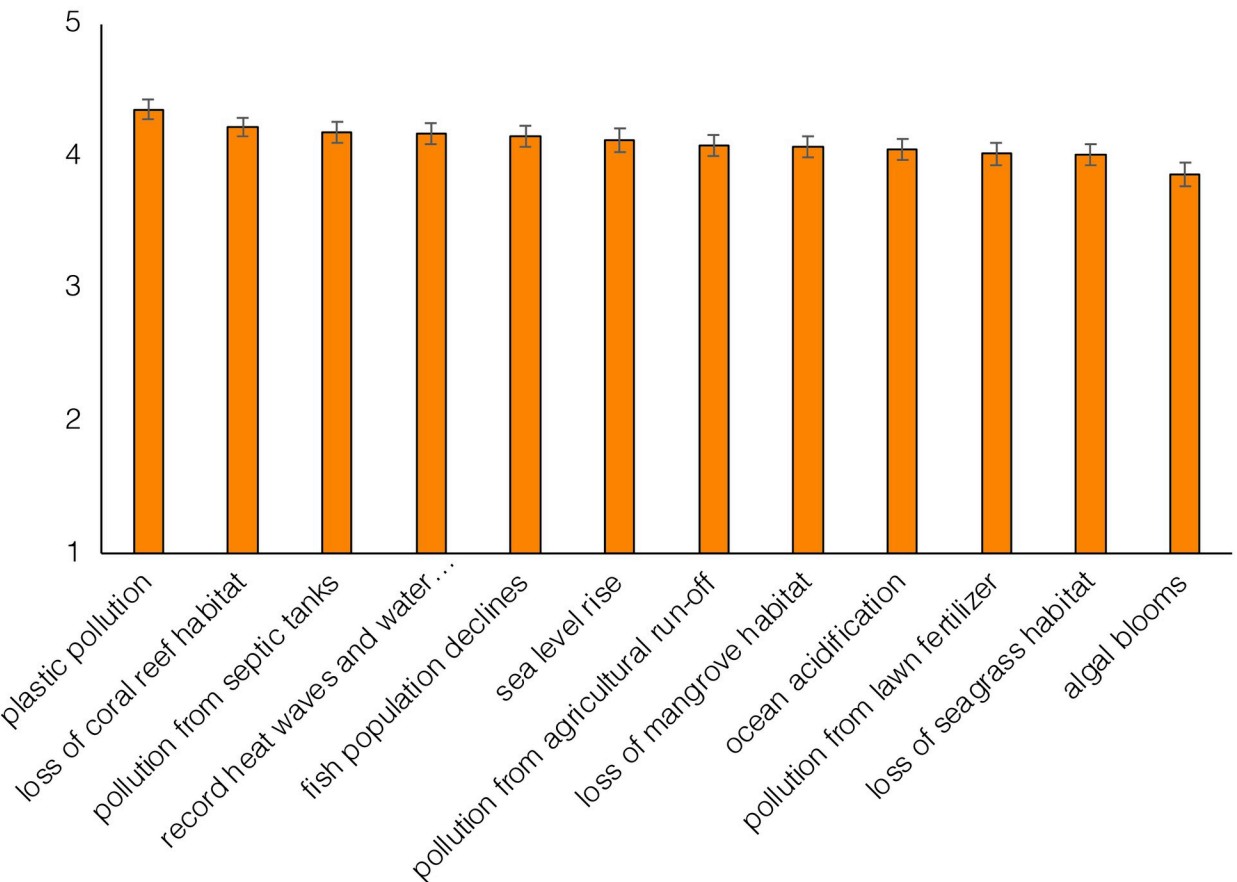

**Fig 3. Relative concern about local environmental threats.** Estimated marginal mean concern associated with local environmental threats on a scale from 1 ("not at all concerned") to 5 ("extremely concerned"). Error bars represent 95% confidence interval.

however, was high. The policy with the lowest level of support still had an average rating of 4.05 on a scale of 5.

All three of the policy scenarios considered for "willingness to pay" were supported by more than 70% of participants even if it meant a higher annual tax rate (Table 4). The proportion of those saying they would support a policy differed across policy treatments ($\chi^2(2) = 9.34$, $p < 0.01$, $\varphi = 0.09$). Post-hoc z-tests with a bonferroni correction found significantly less support ($p < 0.05$) for septic tank conversion than for wastewater treatment infrastructure, with living shoreline restoration falling in between and not differing significantly from other policy treatment options.

The highest amount participants reported being willing to pay did not significantly differ across policy options ($H(2) = 0.11$, p = 0.95). The median highest willingness to pay across all policies was $100.00 (USD) in annual taxes (IQR $50.00—$600.00). For those that reported that they were unwilling to support the proposed policies, the most frequently selected reasons were as follows: a) "I already pay enough taxes" ($n = 132$), b) "I think the government should do this without raising taxes" ($n = 86$), c) "I'm not sure the money would be put to good use" ($n = 75$), d) "I couldn't afford to pay an increase in taxes ($n = 66$), and e) "This environmental resource is not worth an increase in taxes ($n = 22$). A further 23 participants selected "I do not know/no answer".

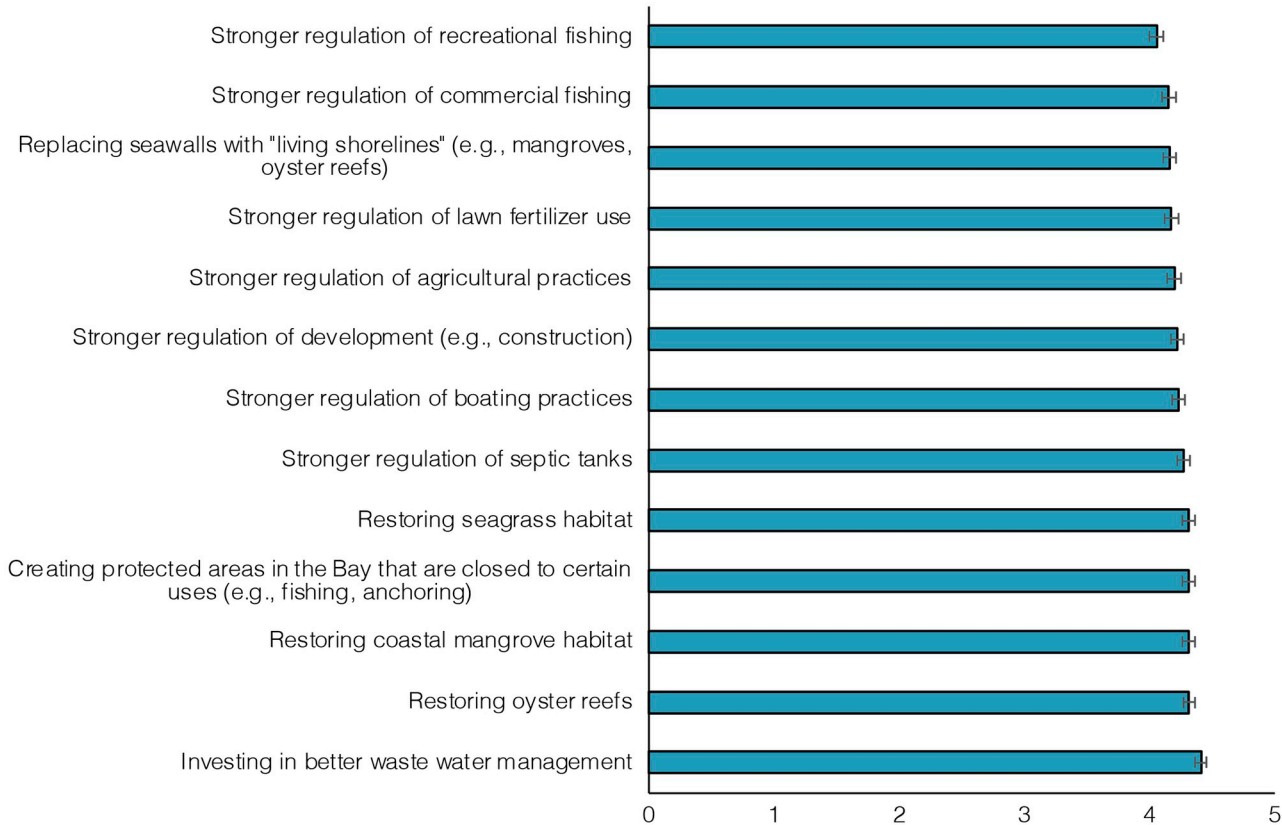

**Fig 4. Support for policy actions.** Estimated marginal mean support for policy actions on a scale from 1 ("strongly do not support") to 5 ("strongly support"). Error bars represent 95% confidence interval.

## Perception of local government

Participants were given an "I do not know" option when rating items about County government performance. Across the five relevant items, between 9.4% and 14.1% of participants selected this option. The remaining participants generally endorsed positively valenced statements about the County government's management of Biscayne Bay. When compared, participants agreed more with statements about the County's commitment and ability to protect the Bay than they did with statements about how well or fairly they were completing the job ($F$ (3.9) = 20.62, $p < 0.001$) (Fig 5).

**Table 4. Number and percent of participants who said they would support each policy even if it raised their annual taxes.**

| Policy scenario | Total $n$ in each group | "yes" | "no" |
|---|---|---|---|
| | | n (%) | |
| Septic tank conversion | 372 | 271 (72.8%) | 101 (27.2%) |
| Wastewater treatment infrastructure | 366 | 301 (82.2%) | 65 (17.8%) |
| Living shoreline restoration | 366 | 284 (77.6%) | 82 (22.4%) |

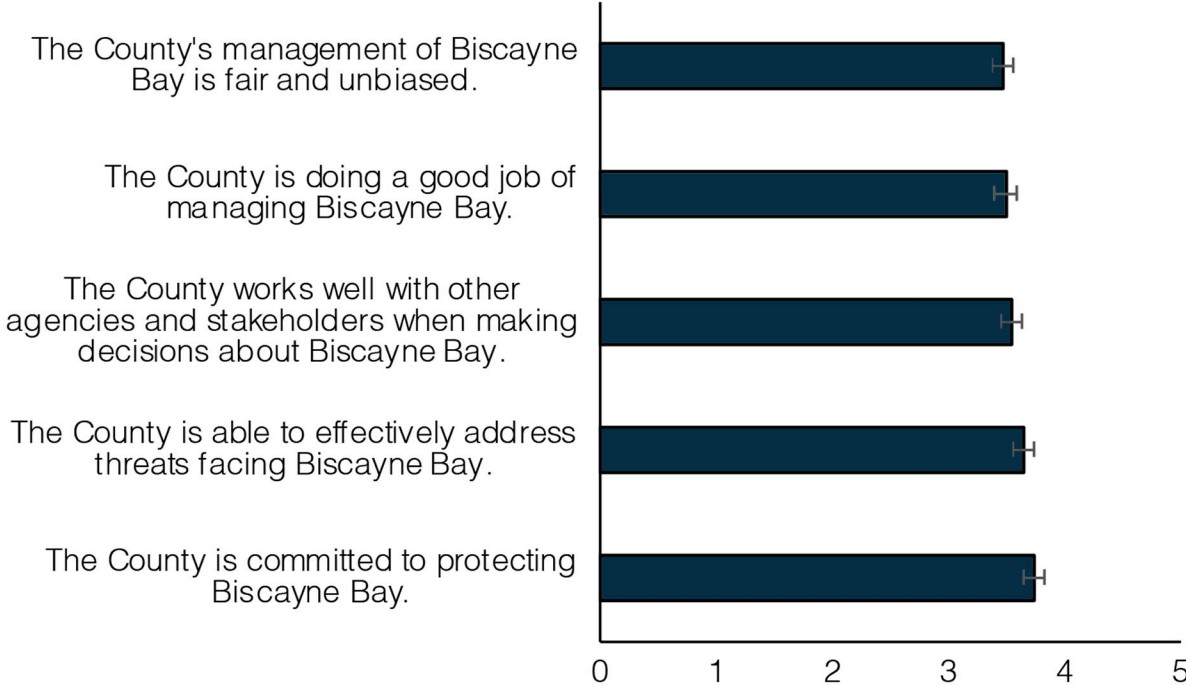

**Fig 5. Perceptions of local governance.** Estimated marginal mean response to value statements associated with the Miami-Dade County government on a scale from 1 ("strongly disagree") to 5 ("strongly agree"). Error bars represent 95% confidence interval.

### Individual differences and key outcomes

The association of several key outcome variables with individual difference factors are presented in Table 5. The outcome variables concern, values, policy support, and county perceptions represent aggregate scores from multi-item scales ($\alpha > 0.7$ for all scales).

## Discussion

In this study, I present the first comprehensive exploration of public stakeholder perceptions, values, attitudes and interactions in the Miami-Biscayne Bay social-ecological system. Results describe a public that interacts with and utilizes Biscayne Bay in a variety of ways, from leisure and recreation to subsistence. This public believes the Bay to be moderately healthy, though somewhat in decline, and has experienced a range of local environmental threats, about which they feel considerable concern. These interactions and concerns are in turn reflected in overwhelming endorsement of value statements regarding the ecological, material, cultural and economic importance of the ecosystem to the city, as well as high levels of support for policy actions to protect and restore that ecosystem. Together these findings indicate that additional policy steps to preserve and restore Biscayne Bay would enjoy support from the local public.

### Public interactions with and use of Biscayne Bay

First, our results indicate that the people of Miami interact with and utilize Biscayne Bay regularly and in a variety of ways. The overwhelming majority visited the Bay at least once per year. The frequency and popularity of certain activities, including beach visits and recreational angling were not surprising—beach tourism and recreational angling are both extremely popular statewide and considered important local industries [59, 60]. Fishing for food was the

**Table 5. How were individual differences associated with key outcome variables.**

| Individual differences | Outcome variables | | | | | |
|---|---|---|---|---|---|---|
| | *Perceived state* | *Perceived trend* | *Concern* | *Values* | *Policy support* | *County perceptions (positive)* |
| **Groups** | *F (df)—Eta* | | | | | |
| **Race** | 11.89(3) - 0.04*** | 7.83(3) - 0.03*** | nd | nd | nd | 5.24(3) - 0.02** |
| *Asian* | 2.5 | 2.43 | - | - | - | 3.34 |
| *Black or African American* | 3.29 | 3.21 | - | - | - | 3.9 |
| *White* | 2.62 | 2.73 | - | - | - | 3.53 |
| *Other* | 2.6 | 2.73 | - | - | - | 3.36 |
| **Political party** | 9.20(2) - 0.03*** | 10.13(2) - 0.03*** | 4.81(2) - 0.01** | 3.73(2) - 0.01* | 6.15(2) - 0.01** | 13.29(2) - 0.05*** |
| | *Mean* | | | | | |
| *Republican* | 3.03 | 2.93 | 4 | 4.45 | 4.23 | 3.76 |
| *Democrat* | 2.99 | 2.95 | 4.16 | 4.55 | 4.36 | 3.76 |
| *Independent* | 2.55 | 2.42 | 3.98 | 4.42 | 4.49 | 3.23 |
| | *t-test—Cohen's d* | | | | | |
| **Ethnicity** | nd | nd | 2.21 - 0.14 | nd | nd | nd |
| *Hispanic or Latino* | - | - | 4.1 | - | - | - |
| *Non-Hispanic or Latino* | - | - | 3.98 | - | - | - |
| **Gender** | 5.06*** - 0.36 | 5.02*** - 0.36 | -2.08* - 0.13 | -2.02* - 0.12 | nd | nd |
| *Male identifying* | 3.05 | 3.00 | 3.99 | 4.43 | - | - |
| *Female identifying* | 2.63 | 2.53 | 4.10 | 4.51 | - | - |
| **Rent vs own** | nd | nd | nd | nd | nd | nd |
| *Rent* | - | - | - | - | - | - |
| *Own* | - | - | - | - | - | - |
| | *Spearman Rho* | | | | | |
| **Age** | -0.17*** | -0.18** | ns | 0.14*** | ns | -.09* |
| **Income** | 0.17*** | 0.10*** | 0.11*** | 0.08** | 0.13*** | 0.17*** |
| **Highest level of education** | 0.21*** | 0.10** | 0.08** | 0.12*** | 0.18*** | 0.20*** |
| **Liberal ideology** | ns | 0.10** | 0.22*** | 0.17*** | 0.23*** | 0.16*** |
| **Years of residence in Miami** | -0.17*** | -0.20*** | ns | ns | ns | -0.16*** |

second most common activity amongst participants. The importance of recreational and commercial angling for coastal management are well recognized in the scientific literature [61–65]. This finding, however, indicates a separate and important component to how the public is interacting with local fisheries. Urban subsistence fishing has been noted elsewhere to be relatively invisible to both management and science at present [66]. These activities and the spaces they inhabit can be important for community health and well-being, including but not limited to food security [67]. They can also be subject to conflict with other coastal users and associated with shifting access and legality [67]. As such, much more research is needed to understand both the potential ecological impacts and social roles of urban food fishing.

Prior research has explored differing attitudes between stakeholder groups, including those that identify with particular activities or practices, (e.g., divers, fishers). When constructed as distinct groups, patterns emerge in terms of attitudes toward problems and policies, and views of potential trade-offs [68, 69]. My findings, however, highlight the degree to which, in the general public, these identities are not clear cut—with most participating in multiple kinds of activities every year. It is possible that policy forums may produce more identity-based discourse (e.g., speaking as a "diver" or "fishermen") by attracting or selecting for those with stronger identities or by encouraging identity-salience through participatory structures [e.g., 70].

Public interaction with natural environments is a complex management concern. On the one hand, experiences in nature and natural resource use is associated with caring for and valuing those resources [71, 72]. On the other, resources users may conflict with each other, and can also negatively impact the ecosystem itself (i.e., through litter, boat propeller scars, overfishing) [73–77]. In Biscayne Bay, the inherent trade-offs between public access to resources, economic productivity, and environmental protection have been acknowledged for years. Management of those trade-offs will require consistent and engaged governance. In Miami, these trade-offs are exasperated by issues of equity, justice, and development. While Biscayne Bay is managed as an open-access commons, in reality, much of the coastline has been privatized for decades [53]. When combined with the lack of public transportation infrastructure in the County, the closure and privatization of coastal spaces has resulted in local environmental injustices—access to environmental amenities including beaches is limited for residents in proportionally non-White, socioeconomically disadvantaged neighborhoods [54].

## Perceived Biscayne Bay value, quality and ecosystem trend

On average, participants believed Biscayne Bay to be moderately healthy and to have only slightly declined in the last decade. This assessment contrasts with descriptions in recent government research reports which describe the ecosystem as being "at a tipping point" [78, p. 3] and in danger of an ecosystem "regime shift" [79, p. 1799]. Considered positively, this finding could indicate that the public still recognizes the ecological value of the Bay, which may relate to support for protection. However, the public perception literature suggests potential problems.

For instance, declines in marine ecosystems may be less perceptible compared with those in terrestrial ecosystems. This can make marine ecosystems especially vulnerable to "shifting baselines syndrome", in which the perceived natural or healthy state of an ecosystem (the "baseline"), becomes increasingly worse as successive generations grow up accustomed to increasingly damaged habitats (the "shift") [80, 81]. A shifting baselines explanation is supported by the finding that age correlated with negative perception of both the state and general trend of Biscayne Bay. Previous research also indicates that major declines in southeastern estuaries may have occurred more than two decades ago (e.g., of bonefish in the 1990s, of reef fish assemblages prior to 2006) [82, 83]. Ultimately, a biased perception of ecological baselines arise from a lack of scientific baseline data or personal interactions with ecosystems, or both, and can impact management decision-making and goal-setting [84].

Despite any perceived declines, the public attributed a variety of values to Biscayne Bay including economic, ecological, cultural, and existential importance. Prior environmental psychology research has identified values as particularly important for explaining policy support (e.g., Values-Beliefs-Norms model) [85–87]. While economic arguments for environmental protection can dominate discourse in conservation and governance spaces [4], in this study, economic value was rated relatively lower than other kinds of values. While this does not mean that economic arguments do not belong in conservation and policy messaging, it may indicate that other kinds of messaging are being left on the table. Diverse social and cultural values are critical to marine spatial planning, but the difficulty in measuring and quantifying such values can mean they are less readily incorporated into policy decision-making [88]. In contrast, many more abstract cultural and experiential values are more readily understood, described, and experienced by the public as they are tied to lived daily experience [8, 89]. Overall, these findings highlight that Biscayne Bay remains an important cultural symbol amongst the local public [53].

## Experienced threats and concern

Overall, rates of personal experience with and concern about local environmental threats were high in this sample. The public in Miami are exposed to, aware of, and worried about a variety of local environmental problems. Findings suggest a relationship between experience, threat visibility and concern. Plastic pollution was the threat participants reported most commonly experiencing and being impacted by, and was also rated as the most concerning. The high level of salience and concern associated with plastics is echoed in other studies from Florida [90, 91], and around the world [e.g., 7]. One possible explanation for this is the high visibility of plastic, at least before it breaks down into its more insidious microplastic form [92, 93].

In contrast, algal blooms were the least commonly noted threat amongst participants, with less than one fifth saying they had personally observed or been affected by a bloom. Similarly, participants reported being significantly less concerned about algal blooms than all other listed threats. Again, visibility, combined with shifting baselines may explain this pattern. Florida is impacted by several types of harmful algal blooms [94]. Red tide (*Karenia brevis)* has received considerable media and research attention and is associated with fish kills, marine megafauna death, and a variety of human health impacts, and is most common on the west coast of the State [94, 95]. In contrast, Biscayne Bay largely suffers from a persistent green macroalgae bloom (*Anadyomene sp*.) which dominates and outcompetes the seagrass beds which form the foundation for the local food chain [96, 97]. These blooms are associated with nutrient pollution which has slowly altered local marine ecosystems, in some cases resulting in localized regime shifts. These blooms only sometimes culminate in anoxic events and corresponding fish kills, which due to water exchange and local geography, tend to concentrate most visibly in the northern parts of Biscayne Bay [49, 98, 99]. Thus while nutrient pollution is a fundamental and persistent issue in the bay, the relatively slow and hidden nature of resulting impacts may be hindering proportional public awareness and concern.

Relative concern over different habitats is also of note here. In this study, participants rated loss of coral habitat as more concerning than mangroves or seagrass. This tracks with other studies in Florida which found higher support for coral restoration efforts than seagrass or mangrove restoration [91]. Duarte [100] showed that a bias for relatively "charismatic" coral reef ecosystems relative to other coastal habitats is found in media coverage and research effort. Others have argued that this bias and associated public attitudes has important consequences for policy and management [101, 102]. Despite being discussed, and at times managed separately, coastal habitats are intricately linked and interdependent [103–106]. From a coastal conservation perspective, differential public concern and awareness is important to note as outreach, education, and policy messaging may need to make these fundamental connections visible and evident to the public. Despite differences noted in this section however, concern over environmental threats was overall very high in this survey, with important implications for policy support.

## Policy support and key related variables

All of the potential policy actions presented were rated highly favorably by participants. Policy support was high across a variety of potential options even when it would involve an increase in annual taxes. In both the Likert-type scales and willingness-to-pay questions, the highest levels of support were granted to ecosystem restoration efforts and investment in wastewater management. High levels of public support for restoration has been found in other research including in Florida [91] and Europe [107]. Conversely, septic tank conversion, which is primarily associated with private residences in particular communities, can be more contentious (i.e., with regard to who is required to bear the costs of conversion or maintenance when

harms are mostly felt in the commons) [108]. Other findings however indicate that underlying cause-effect relationships and potential trade-offs associated with habitat restoration may not be fully understood. Restoration efforts that do not address underlying causes of degradation for instance often fail [109].

Similarly, while mangrove and oyster reef restoration were highly supported in this survey, replacing seawalls with living shorelines (including mangroves) and regulating construction and development received slightly, but significantly less support. These findings may reflect inherent trade-offs between private land-owners and other stakeholders, and between prioritizing economic growth, public bay access, shoreline protection, and environmental protection. In other studies, shoreline renaturalization was supported, but the relationship between climate impacts, public access, and shoreline management decisions was poorly understood [110]. Conservation messaging will need to clearly communicate about holistic and diverse shoreline strategies, potential trade-offs, and potential benefits (e.g., fish population health and property protections offered by mangroves) [111, 112].

Policy support was relatively lower (but still high) for new potential regulations, particularly of fishing. Recreational fishing in particular is both popular and economically important to the State [113]. While fishing regulation was less supported, area closures or marine protected areas were supported highly. Despite this pattern of public policy preference, it is noteworthy that recent attempts to create a small area closure in Biscayne National Park was met with significant, organized opposition and was considered controversial in local media coverage [114]. Ultimately, managing agencies opted for new fishery regulations instead of the closure, which one study found was not representative of local angler attitudes [115]. These findings in this context indicate that public support for particular policies and those advocated for by professionalized stakeholder groups, may not always align. In reality, the specific structures of participatory processes—who is included in policy discussions and how—will have significant impacts on policy selection and legitimacy [116, 117].

Overall, however, it is the high levels of support across all suggested policy actions that is most noteworthy in this study. Public preferences and support for policy are critical for policy success at every stage [118]. High-levels of support make it easier to get an issue on the political agenda, to rally voter bloc action, and to fundraise [118]. Once a policy is passed, high levels of support also helps ensure that policies are viewed as legitimate, which can help smooth implementation and public cooperation [119, 120]. Importantly, in this study, not only was support for policy action high, but differences between key demographics were minor or non-existent. There was no difference in level of support for policy action between racial, ethnic, or gender identities. Similarly, there was no difference in support between those that rent or own their homes in Miami. While support was positively correlated with income, education, and liberal ideology, there was no correlation with age or years of residence.

Finally, a significant difference emerged based on political party identification, but the effect size of this difference was small—with the group with the highest (Independents, $m = 4.49$) and the lowest scores (Republicans, $m = 4.23$) both still decidedly supporting all policies on average. This is notable given the well-documented relationship between ideology and environmental attitudes, and the increasingly partisan nature of environmental policies in the US and globally [121–126].

Policy support does not exist in a vacuum and is often related how the implementing government body is perceived. Both the overall perceived quality and trustworthiness of relevant government institutions are predictive of policy support in context [118]. Public trust in government is composed of both whether one believes an institution will be fair and whether they are viewed as capable of carrying out proposed plans [127, 128]. On these metrics, the Miami-Dade County government was perceived largely positively. Participants however were more

likely to say they felt the County was capable of and committed to protecting Biscayne Bay, than they were to endorse the job the County is currently doing in this task. In aggregate, this study suggests that the local government is in a relatively strong position to act to conserve and restore Biscayne Bay.

## Conclusions

This paper joins a small, but growing body of literature on public perceptions of marine environments. Human population growth continues to concentrate along coastlines. The fundamental importance of marine ecosystems to human economies, culture, and well-being is increasingly recognized. Miami-Dade County is on the forefront of many issues that are emerging at the intersection of coastal marine ecosystems and human society. As such, this study provides critical insight into public perceptions and related management and outreach strategies both locally and for coastal cities around the globe.

Social-ecological systems are inherently complex and the urban-coastal system explored here is no different. Despite this complexity, this study demonstrates the power of public perceptions research to identify gaps and opportunities for management and outreach. In particular, this study highlights the need to address the visibility of impacts, connections, and practices. How and whether the public perceives threats or cause-effect relationships within ecosystems can shape policy preferences, which in turn will impact policy outcomes. Despite the diversity of backgrounds and interactions with local natural resources represented in the study sample, findings also highlight the potential for public perception research to identify opportunities for policy action, particularly at a local scale. In aggregate, this study suggests that the local government is in a relatively strong position to act to conserve and restore Biscayne Bay with broad public buy-in.

## Author Contributions

**Conceptualization:** Julia Wester.

**Data curation:** Julia Wester.

**Formal analysis:** Julia Wester.

**Funding acquisition:** Julia Wester.

**Investigation:** Julia Wester.

**Methodology:** Julia Wester.

**Project administration:** Julia Wester.

**Resources:** Julia Wester.

**Supervision:** Julia Wester.

**Validation:** Julia Wester.

**Visualization:** Julia Wester.

**Writing – original draft:** Julia Wester.

**Writing – review & editing:** Julia Wester.

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
