## [Decision Letter · Decision Letter 0]

25 May 2023

PONE-D-23-07915Public perception of an important urban estuary: Values, attitudes, and policy support in the Biscayne Bay-Miami Social Ecological SystemPLOS ONE

Dear Dr. Wester,

Thank you for submitting your manuscript to PLOS ONE. After careful consideration, we feel that it has merit but does not fully meet PLOS ONE’s publication criteria as it currently stands. Therefore, we invite you to submit a revised version of the manuscript that addresses the points raised during the review process.

ACADEMIC EDITOR: 

 Authors are advised to revise whole manuscript for langauage and i would suggest to take help from native english speaker. And also for gramatical and phrases mistakes. 

We look forward to receiving your revised manuscript.

Kind regards,

Dharmendra Kumar Meena

Academic Editor

PLOS ONE

Additional Editor Comments:

Author are advised to focus on language and grammatical and text rearrangement for making a continuous flow for easy reading

Reviewers' comments:

Reviewer's Responses to Questions

**Comments to the Author**

1. Is the manuscript technically sound, and do the data support the conclusions?

Reviewer #1: Yes

Reviewer #2: Yes

2. Has the statistical analysis been performed appropriately and rigorously? 

Reviewer #1: Yes

Reviewer #2: Yes

3. Have the authors made all data underlying the findings in their manuscript fully available?

Reviewer #1: Yes

Reviewer #2: No

4. Is the manuscript presented in an intelligible fashion and written in standard English?

Reviewer #1: Yes

Reviewer #2: Yes

5. Review Comments to the Author

Reviewer #1: Abstract:

1. Re-write abstract- with brief introduction, importance of study, brief methodologies adopted, salient achievements and

conclusion.

Conclusion:

1. Suggested that author added references in manuscript conclusion which is not required to add references in conclusion. I suggested that re-write the conclusion in brief-way with suggestions.

Reviewer #2: The study on Biscayne Bay shows the multiple parameters including the ecological interaction as well as physical interaction towards the water body by the public. It can be use for multiple purposes if the management practices adopted in a sustainable manners. Some where coma (,) is use with the word and like as (, and) so please correct it.

6. PLOS authors have the option to publish the peer review history of their article (what does this mean?). If published, this will include your full peer review and any attached files.

Reviewer #1: No

Reviewer #2: **Yes: **Dr. Veerendra Singh

---

## [Author Response · Author response to Decision Letter 0]

9 Jun 2023

Response to reviewers

I thank the reviewers and editor for their support and helpful comments on this manuscript. I have carefully reviewed the manuscript and incorporated suggested changes. Point-by-point responses are included in-line below in bold.

Kind regards,

Julia Wester

 The manuscript has been reviewed for formatting requirements. 

The following details have been added to the methods: “At the start of the survey participants were presented with information about the study and provided written consent in the form of checking a box to proceed before they were able to begin the survey.”

Data has been made available via Dryad at the following location: https://doi.org/10.5061/dryad.1rn8pk106

No changes have been made to the reference section. No retracted papers have been cited. 

Author are advised to focus on language and grammatical and text rearrangement for making a continuous flow for easy reading

I have reread and lightly edited the manuscript for flow and clarity.

Reviewer #1: Abstract:

1. Re-write abstract- with brief introduction, importance of study, brief methodologies adopted, salient achievements and conclusion.

I thank the reviewer for their support of the manuscript and helpful comments here. I have revisited the abstract and believe that each of the components listed have been included already. 

Conclusion:

1. Suggested that author added references in manuscript conclusion which is not required to add references in conclusion. I suggested that re-write the conclusion in brief-way with suggestions.

 References have been removed from the conclusions. 

Reviewer #2: The study on Biscayne Bay shows the multiple parameters including the ecological interaction as well as physical interaction towards the water body by the public. It can be use for multiple purposes if the management practices adopted in a sustainable manners. Some where coma (,) is use with the word and like as (, and) so please correct it.

We thank the reviewer for their support of the manuscript. I have reread and lightly edited the manuscript for grammatical or syntactical errors.

---

## [Editor Report · Decision Letter 1]

15 Jun 2023

Public perception of an important urban estuary: Values, attitudes, and policy support in the Biscayne Bay-Miami Social Ecological System

PONE-D-23-07915R1

Dear Dr. Webster

We’re pleased to inform you that your manuscript has been judged scientifically suitable for publication and will be formally accepted for publication once it meets all outstanding technical requirements.

Kind regards,

Dharmendra Kumar Meena

Academic Editor

PLOS ONE

Additional Editor Comments (optional):

Article is recommended for publication
---

## [Editor Report · Acceptance letter]

21 Jun 2023

PONE-D-23-07915R1 

Public perception of an important urban estuary: Values, attitudes, and policy support in the Biscayne Bay-Miami Social Ecological System 

Dear Dr. Wester:

I'm pleased to inform you that your manuscript has been deemed suitable for publication in PLOS ONE. Congratulations! Your manuscript is now with our production department. 

Kind regards, 

on behalf of

Dr. Dharmendra Kumar Meena 

Academic Editor

PLOS ONE